**Data Availability Statement:** All relevant data are within the paper and its Supporting Information files.

# Demographic profiles of contact lens wearers and their association with lens wear characteristics in Trinidad and Tobago: A retrospective study

Ngozika E. Ezinne[1,2☯], Dipesh Bhattarai[3‡], Kingsley K. Ekemiri[1,2], Gabrielle N. Harbajan[1‡], Anesha C. Crooks[1‡], Khathutshelo P. Mashige[2‡], Alex A. Ilechie[4‡], Ferial M. Zeried[5‡], Uchechukwu L. Osuagwu[6☯]*

**1** Optometry Unit, Department of Clinical Surgical Sciences, Faculty of Medical Sciences, University of the West Indies, St Augustine, Trinidad and Tobago, **2** African Vision Research Institute (AVRI), Discipline of Optometry, School of Health Sciences, University of KwaZulu-Natal, Durban, South Africa, **3** School of Medicine, Faculty of Health, Deakin University, Geelong, Victoria, Australia, **4** Department of Optometry and Vision Science, University of Cape Coast, Cape Coast, Ghana, **5** Department of Optometry & Vision Sciences College of Applied Medical Sciences, King Saud University, Ilesha, Saudi Arabia, **6** Translational Health Research Institute (THRI), School of Medicine, Western Sydney University, Campbelltown, Australia

☯ These authors contributed equally to this work.
‡ DB, GNH, ACC, KPM, AAI and FMZ also contributed equally to this work
* L.Osuagwu@westernsydney.edu.au

## Abstract

### Purpose

**C**ontact lenses (CL) remain a popular mode of refractive error correction globally and in the Caribbean, mostly among young people. However, no data on the characteristics of the CL population wearers in the Caribbean is available. This study reported on the characteristics of CL wearers and the associated factors in Trinidad and Tobago (T&T).

### Methods

This retrospective study reviewed the clinical records of 243 CL wearers who attended the University of the West Indies (UWI) optometry clinic between 2017 and 2018. Data on their demographic profiles, CL wearing characteristics including lens type, material, purpose of wear, replacement schedule and lens care systems were extracted and analyzed. The associations between the demographic characteristics and CL wearing characteristics were also determined.

### Results

About half of the CL wearers used them for fashion (more among those aged 18 to 30 years, 61.0%), therapeutic (more among those <18 years, 43.8%, $P = 0.001$) and refractive error correction purposes (more in those >40 years, $P = 0.001$). Females were more likely to use CLs for fashion compared with males (67.0% versus 40.7%). Age ($P < 0.0005$) and gender ($P = 0.030$) were associated with the lens materials. Those aged 18–30 years were more likely use hydrogels compared with the younger ones (64.1% versus 25.0%). Rigid gas

**Funding:** No. The authors had no funding for this study.

**Competing interests:** No. The authors have declared that no competing interests exist.

permeable (RGP) CL use was more common in males than females (21.8% versus 10.9%, $P = 0.031$). Daily disposables were predominantly used by younger respondents (18–30 years old, 31.3%, $P < 0.001$) and more in females than males (66.7% vs 49.4%, $P = 0.040$).

## Conclusion

The study found that the CL appears to be more commonly worn for fashion in younger females and for refractive error correction in older males. Age, gender, and employment status were the main determinants of lens wear among respondents attending the university clinic in T&T.

## Introduction

Contact lenses (CL) are used for the correction of refractive errors as well as fashion and therapeutic purposes [1]. It is estimated that there are over 140 million CL wearers globally [2]. However, over the past decade, the CL industry has experienced advancements in terms of lens designs and materials which has influenced the choice of wear [3–5]. The use of CL has been in existence for the past two decades, with ongoing evaluation of different types of CL worn around the world, to better understand and identify some of the common factors which influence choice of wear. Global trends indicate that soft CL for daily wear is more popular, and majority of wearers were females [6–9]. In addition, an increase in silicone hydrogel CL for daily wear has been reported globally [1, 10, 11]. Rigid CL lens fitting has been shown to represent a small percentage of the market, particularly in developing countries [7, 12]. The common factors reported to affect CL wear are demographic profiles of the population, income, purpose of wear, prevailing eye condition in that area, the range of CL available, level of education of the optometrist and socio-cultural issues [8, 10, 13].

Trinidad and Tobago (T&T) is a small twin Island republic state in the Caribbean [14] with a population of 1.4 million people and is regarded as the power house of the English speaking Caribbean due to its large natural oil and gas reserves [15]. It has only one optometry school which is located at the University of the West Indies, Saint Augustine campus and started in 2011. Prior to 2009, most optometrists in T&T were foreign trained (UK, USA, Nigeria, and South Africa) because there was no optometry training institution in the country and across the Caribbean region. To meet the growing demand for eye care services by increasing the number of optometrists in the region, the first Bachelor of Science programme in Optometry was started in T&T at the University of the West Indies and this training brought a lot of changes in optometry profession across the Caribbean [16].

CL practice in T&T has evolved over the last decades following the emergence of the optometry programme and together with some other countries in the Caribbean including Barbados, Guyana, Saint Lucia and Jamaica, the country has benefitted enormously from this Programme [16] with increase in the optometry manpower. The increase in the number of optometry graduate from the University of West Indies led to the expansion of optometry practice with many of the optometrists embarking on different areas of interests including CL practice. This is possibly due to a high demand for CL wear in private practices because of the improvement in the socio-economic condition of many in the country.

CL practice in T&T is regulated by the Trinidad and Tobago Optometrists Association (TTOA) [17]. Optometrists and Ophthalmologists are the only professionals permitted to prescribe CL for the management of refractive error, keratoconus and anisometropia in T&T. The use of cosmetic CL is not regulated as individuals are free to purchase them from beauty parlor, over-the-counter pharmacy shops, online and even supermarkets. Most of the big CL

companies such as Johnson and Johnson, Acuvue, Bausch and Lomb have their representatives in the country making various CLs readily available.

CL are gaining popularity worldwide as an alternative to spectacles for the correction of refractive errors and across different countries, their usage varies with various demographic characteristics [5, 6, 8, 9]. Approximately 7.2%, 12.0%, 14.6% and 14.7% adult CL usage rates were reported in the UK, Norway, Japan, and the USA respectively [11]. Unlike the neighboring countries like the United States, there is a paucity of published data on CL use across the Caribbean and the characteristics of lens wearers in T&T remain unknown. Considering the growing popularity of CL use and the fact that demographic profile of lens wearers and associated factors might vary between countries due to the variations in socioeconomic factors, more studies are needed to provide an insight into CL wear in different countries including T&T. The information will aid eye care practitioners and lens manufacturers in T&T to understand the local CL market to assess demand, growth in relation to other countries. The aim of this study was to determine the demographic characteristic of CL wearers and associated factors among respondents who attended a university optometry clinic in T&T. This will provide data for comparison with the global findings.

## Materials and methods

### Study setting

This study was carried out at the University of the West Indies (UWI) optometry clinic situated in Couva, an urban town (48,858 in 2011 census) in West-central Trinidad, south of Port of Spain and Chaguanas [18]. It is the capital and main urban centre of Couva–Tabaquite–Talparo located directly adjacent to the Venezuelan state of Monagas [18]. It is the fourth least populated city in T&T and has very few optometry clinics. Most (80%) of the optometry practices in T&T are situated in Port of Spain, which is the nation capital, and it can be assumed that most CL wearers live there.

The UWI optometry clinic in T&T provides optometry training and services to the public. It is well equipped and has a highly qualified academic staff complement. CL practice including fitting and prescribing CL for keratoconus management is among the eye care services provided.

### Ethical considerations

Approval to conduct the study was obtained from the University of the West Indies, St Augustine campus, Research and Ethics Committee (registration number: CREC.SA.0037/09/2019). Permission to assess patients' case files was obtained from the university optometry unit Coordinator, however data were fully anonymized before researchers accessed them. The study followed the tenets of the Declaration of Helsinki.

### Study design

This was a retrospective review of data from patients prescribed with CL at the UWI optometry clinic from January 2017 to December 2018. The records of all patients prescribed with CL were retrieved and reviewed.

### Inclusion and exclusion criteria

Data for all patients prescribed with CL at UWI optometry clinic from January 2017 to December 2018 were included. All patients who wore CL prescribed outside UWI optometry clinic and those whose CLs prescriptions and details could not be verified, were excluded.

### Data collection procedure

Data collection involved the use of a data extraction sheet to obtain information on the demographic profile of each patient fitted and CL wearing history and characteristics. These were based on a review of previous studies [1, 6, 7] and included lens type, purpose of wear (fashion [i.e. those that are used for masking eye flaws and or enhancing appearance], refractive correction and therapeutic [i.e. CLs used for the purpose of managing cases such as aphakia, keratoconus, irregular cornea, amblyopia therapy and conditions such as ocular surface problems which need specialized lenses for their management]), [9, 13, 19, 20] lens material (hydrogel, silicon hydrogel and rigid gas permeable), habitual lens replacement schedule (daily disposable, weekly disposable, monthly disposable and yearly disposable) [1], lens wear modality (fulltime wear was considered as ≥8hrs per day and seven days a week and part time as <8hrs per day) [7], and lens care systems including lens solutions advised for each patient, over two years.

### Statistical analysis

Data were analysed using the IBM Statistical package for Social Sciences (SPSS) version 27 (SPSS Inc., Chicago, USA). Descriptive statistics were computed for all variables after the data had been screened and the assessed for normality of distribution. Continuous and discrete variables were analyzed using student $t$ test for means and Pearson's Chi-square test for proportions, respectively.

## Results

### Demographics of contact lens wearers

Table 1 presents the demographic profiles and CL-related characteristics. Two hundred and forty-three patients were prescribed with CL at the UWI optometry clinic from January 2017 to December 2018. Their mean age was 29.7 ± 12.7 years (range 4–73 years), and the majority (59.7%) were in the 18 to 30 years age group. Almost two-thirds (64.2%) were females and 54.4% were from urban areas. About half (51.4%) were unemployed and these were mostly young people (64.8% were aged 18–30 years).

### Contact lens wear profile

The main reason for CL wear was fashion (53.5%) followed by refractive correction (27.2%) (Table 1). More than two-thirds (44.9%) wore soft spherical or soft lenses (43.5%). For more than half of the wearers, their CLs materials was polymer-hydrogel (53.1%) and 32.1% wore silicone-hydrogel lenses while (14.8%) wore RGP lenses. Most (60.5%) wore daily disposable CLs and 15.6% wore monthly disposables.

   Most (61%) record cards did not reflect the type of lens care system used. About one-quarter (24.3%) used multipurpose solution [e.gs include Bausch and Lomb Renu advance formula, Renu fresh, Renu sensitive and Biotrue multipurpose solutions used for soft CLs] and 14% used Boston solution by Boston Advance® care system (Bausch & Lomb, Rochester, New York).

### Multipurpose solution

The mean number of the daily hours and the average number of days the wearers used their lenses were recorded as 7.8 (SD, 3.1; range 1.5–18 hours) per day and for about 4 days per week. Majority (60%) wore CL for more than 16 hours per day and 44.45% wore them more than 4 days a week.

**Table 1. Demographic characteristics of the participants and CL-related variables (n = 243 unless otherwise specified).**

| Variables | Number of participants (%) |
|---|---|
| **Year** | |
| 2017 | 101 (41.6) |
| 2018 | 142 (58.4) |
| **Demography** | |
| **Age category** | |
| <18 years | 16 (6.6) |
| 18–30 years | 145 (59.7) |
| 31–40 years | 38 (15.6) |
| >40 years | 44 (18.1) |
| **Gender** | |
| Male | 87 (35.8) |
| Female | 156 (64.2) |
| **Residency (n = 240)** | |
| Rural | 107 (44.6) |
| Urban | 133 (55.4) |
| **Occupation** | |
| Employed | 118 (48.6) |
| Unemployed | 125 (51.4) |
| **Purpose of CL wear (n = 226)** | |
| Fashion | 130 (57.5) |
| Refractive error Correction | 66 (29.2) |
| Therapeutic | 30 (13.3) |
| **CL types (n = 207)** | |
| Soft spherical | 93 (44.9) |
| Soft toric | 90 (43.5) |
| Soft multifocal | 24 (11.6) |
| **CL material** | |
| Hydrogel | 129 (53.1) |
| Silicon hydrogel | 78 (32.1) |
| Rigid Gas permeable | 36 (14.8) |
| **Habitual CL replacement schedule** | |
| Daily disposable | 147 (60.5) |
| Biweekly disposable | 22 (9.1) |
| Monthly disposable | 38 (15.6) |
| Yearly disposable | 36 (14.8) |
| **CL care system** | |
| Multipurpose solution | 59 (24.3) |
| Boston solution | 34 (14.0) |
| None (not recorded) | 150 (61.7) |
| **CL wear modality (n = 200)** | |
| Part time (<8hrs/day) | 80 (40) |
| Full time (≥8hrs/day) | 120 (60) |

CL = contact lens, wear modality. Fashion refers to CL used for masking eye flaws and or enhancing appearance; therapeutic purpose refers to CLs used for the purpose of managing cases such as aphakia, keratoconus, irregular cornea, amblyopia therapy and conditions such as ocular surface problems which need specialized lenses for their management. Boston solution = Boston Advance® care system (Bausch & Lomb, Rochester, New York).

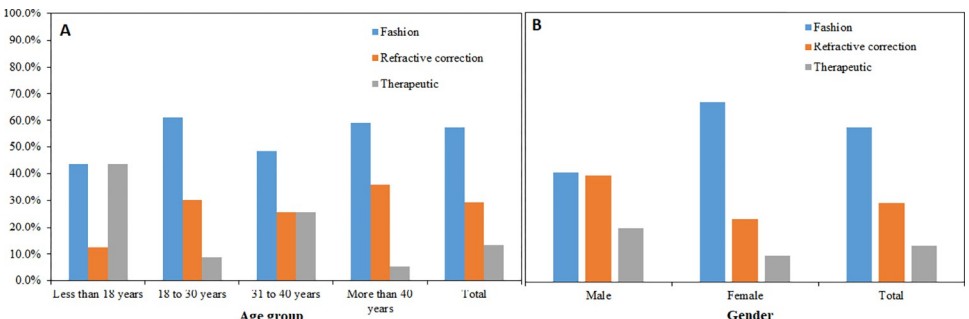

**Fig 1.** Distribution of purpose of contact lens wear by: (A) age group and (B), gender of wearers.

## Association between demographic variables and the purpose of wear

Fig 1 shows the results for the purpose of CL wear by a) age group and b) gender. Chi-square analysis showed that the purpose of CLs wear differed between age groups ($P = 0.001$) and gender of the CL wearers ($p = 0.001$). Compared with other age groups, those in the 18 to 30 years age group were more likely to wear fashion CLs (61.0%) whereas therapeutic purposes were the main reason for CL use among those younger than 18 years (43.8%, $P = 0.001$). In contrast, CL wearers older than 40 years (35.9%) were more likely to use CLs for refractive error correction compared to younger wearers (<18 years, 12.5%, $P = 0.001$).

A higher percentage of females (67.0% versus 40.7%) wore CL for fashion reasons whereas more males than females (39.5% versus 23.4%) used CLs for refractive error correction ($P = 0.001$).

## Association between demographic variables and the CLs material

Age and gender were significantly associated with lens materials worn (both P-values <0.05) (Fig 2). Participants between the ages of 18 and 30 years were more likely to be prescribed hydrogel CLs compared with teenagers (less than 18 years) (64.1% versus 25.0%, $P < 0.001$) and older adults (more than 40 years) (34.1%, $P < 0.001$), who were more likely to be prescribed silicon hydrogel CL. There was a significant association between CL material and gender, and 58.3% of females compared with 43.7% of males were prescribed hydrogel CLs ($P = 0.031$). No other significant association was found between the material prescribed and other demographic variables.

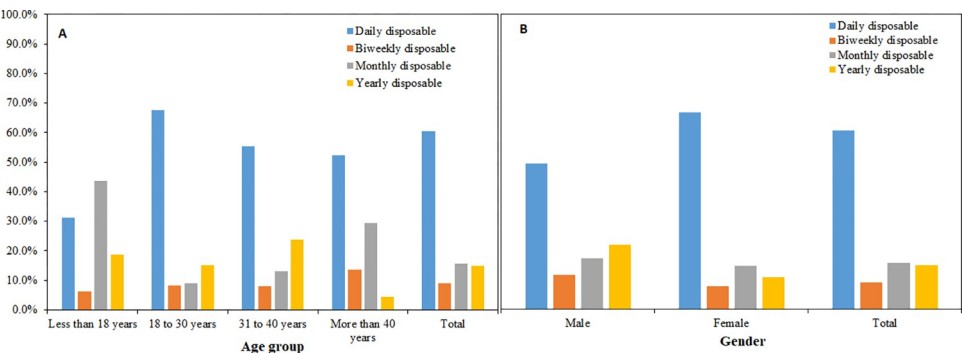

**Fig 2.** Distribution of purpose of contact lens materials by: (A) age group and (B), gender of wearers.

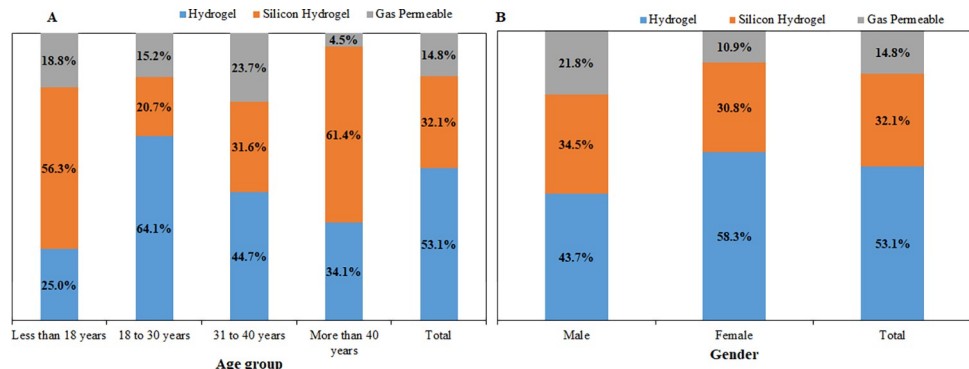

**Fig 3.** Distribution of contact lens wear regimen/replacement schedule *by*: *(A)* age *group* and *(B)*, gender of wearers.

## Association between demographic variables and the replacement schedule (wearing regimen) of CLs

Age and gender were associated with the CL wear regimen ($P < 0.001$) (Fig 3). A higher proportion of those below 18 years of age used monthly disposable CLs (43.8%) compared with other age groups (43.8% versus maximum of 29.5% among those aged more than 40 years, respectively, $P < 0.001$). Compared with other age groups, teenagers were less likely to use daily disposable CLs, which was rather more commonly used by females than males (66.7% versus 49.4%). By contrast, males were more likely to use extended wear CLs (yearly disposables, 21.8% versus 10.9%, $P = 0.040$) than females.

## Association between demographic variables and the modality of wear

The modality of wear (full time or part time wear) was not dependent on the age group ($P = 0.068$) and gender ($P = 0.718$). However, in those older than 18 years, there was a tendency for more wearers to use their lenses for eight or more hours each day (46.2% in the <18 years age group compared with 55.4%, 75% and 70.6% in the 18–30, 31–40, >40 years age groups, respectively). Majority of those who were employed wore their lenses for about eight or more daily hours compared with those who were unemployed at the time of this study (68.0% versus 52.4%, $P = 0.030$).

## Association of lens care system and wearing modality

Fig 4 shows the distribution of the CLs by wearing modality. Many (71.8%) silicone hydrogel and 2.3% of hydrogel wearers used multipurpose solutions to care for their lenses while 94.4% of RGP CL wearers used the Boston solution. There was a significant association between the lens care systems and the CL wear regimen/replacement schedules ($P<0.0005$). All CL wearers who were prescribed weekly replacement lenses used multipurpose solution. However, lens care system used by some monthly replacements (2.6%) and twice more of the yearly replacement CL wearers (5.6%) were not recorded in their files at the time of the review.

## Discussion

This retrospective study investigated the characteristics of CL wearers and their association with lens characteristics using data from a clinic population in T&T over two years. The study found that CLs wearers were mostly young, unemployed females who lived in the urban areas

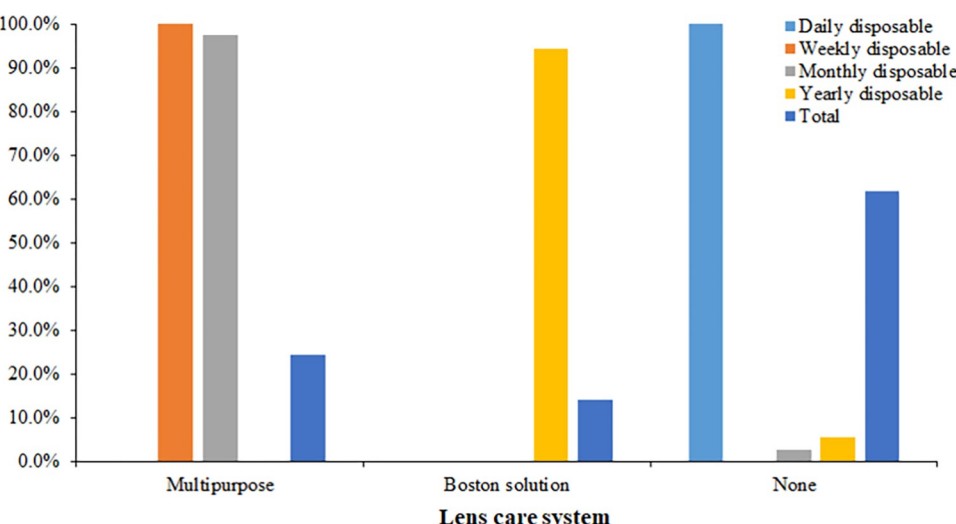

**Fig 4. Distribution of contact lens care systems by wear modality.**

of T&T. Soft, polymer-hydrogel and daily disposable CLs were the most frequently used CL types, materials and wearing regimen, respectively among the CL wearers. Although most CL wearers were full time lens wearers, fashion was cited as the main reason for lens wear.

The findings that most CL wearers were females and used them for fashion reasons are consistent with previous studies and this can be attributed to the perception that CLs are used for fashion [4, 6, 8]. The fact that daily disposable CLs were more common among females than males suggests that females may be prioritizing their expenditure on facial fashion than males since wear of daily disposable lenses is a more expensive option than extended wear lenses [12]. Similarly, cost of daily disposable lenses may explain the finding that daily disposable CLs was not the lens of choice for teenagers.

Most of the CL wearers in our study were between 18–30 years and only few were above 40 years. This is consistent with findings from other studies [7, 21–23] and Morgan et al [24] report of 31.70 (± 14.8) years as the global average age of CL wearers. This could be because this age group is more aware of better optical, occupational, and cosmetic benefit of CL when compared with spectacles. Contrary to our study findings on older age group, international CL prescribing surveys in developed countries including Germany, Canada, Australia, Netherlands, United Kingdom, and United States recorded a higher use of CLs among older age groups (≥30 years) [25, 26]. CL use is still growing in T&T just like in other developing countries. More awareness on the availability of various CLs for older age groups in T&T are highly advised.

Most older CL wearers (> 40 years) in our study used CLs for refractive error correction, which is consistent with trends observed in the USA where the use of CLs for presbyopia correction is on the rise [11]. This may be due to the advancement of CL materials [4] and better understanding of the relationship between the corneal surface and the prescribed CL materials [3]. In contrast, younger patients showed a preference for fashion CLs use, however, 64.8% were unemployed compared with only 13.2% of those aged 31–40 years who also had more people using. While the use of CLs for fashion purpose among young people in this review is high, there is a tendency for increased ocular complications in this age group due to poor compliance [9]. There is need for targeted CL education either in form of booklet or media campaign to encourage proper lens hygiene and improve compliance [27].

Half (51%) of the CL wearers in this study were unemployed and this could be because most of them were students. Proximity of the UWI optometry clinic to the University of the West Indies, Saint Augustine main campus could be the reason for the high utilization of CL among students recorded. High utilization of CLs among students were also reported in studies in Abuja, Nigeria [28] and Sudan [1]. Moreover, students are more conscious of appearance which explains the reason for fashion as the major indication for CL wear in this study.

Previous studies have shown that CL related complications including discomfort were higher among presbyopes than non-presbyopes [13, 19] while young lens wearers showed poorer care/compliance than their older counterparts [9]. In a study done by Young et al., they reported that silicone-hydrogel CLs provided better comfort than the hydrogel CLs, especially in the adverse environment [29]. These may explain the preference to fit most teenagers and presbyopes with silicon-hydrogel CL material compared with most adults aged between 18 and 40 years, who were fitted with hydrogel CLs material in this study.

## Limitations and strengths

The following limitations should be considered in the interpretation of the study findings and when comparing this study to previous or future studies. First, a valid inference could not be made from our data for the whole T & T population since the data was collected from one clinic. Second, due to the retrospective nature of the study, individual's CL wear and care habits could not be investigated. Third, without directly observing the CL wearers, the researchers could not verify the recorded responses in the patients' files. In addition, incomplete documentation common in retrospective studies was another limitation. For example, the type of lens care system used was not recorded for majority of the patients in this study. Despite the limitations, our study provided baseline data on the demographical profile of CL in T&T and could be used for comparison across the Caribbean in future studies. Although, our study sample may be biased towards clinic population it reflected the small population of the island (1.399 million) and greater than the sample size used in similar studies elsewhere including Ghana [8], Australia [30], Malaysia [31], Maldives [32], Saudi Arabia [33], South Africa [34], and Jordan [35]. Further studies on CL use in the Caribbean including the investigation of other factors such as types of refractive error, knowledge and attitude of lens wearers and hygiene are warranted.

## Conclusion

CL use appears to be more common among younger females for fashion and older males for refractive correction. Age, gender, and employment status were the main determinants of lens wear among CL wearers attending the university clinic in T&T.

## Supporting information

**S1 Data.**
(XLSX)

## Author Contributions

**Conceptualization:** Ngozika E. Ezinne.

**Data curation:** Ngozika E. Ezinne, Kingsley K. Ekemiri, Gabrielle N. Harbajan, Anesha C. Crooks.

**Formal analysis:** Dipesh Bhattarai, Uchechukwu L. Osuagwu.

**Funding acquisition:** Ferial M. Zeried.

**Investigation:** Ngozika E. Ezinne, Kingsley K. Ekemiri, Gabrielle N. Harbajan, Anesha C. Crooks, Khathutshelo P. Mashige, Uchechukwu L. Osuagwu.

**Methodology:** Ngozika E. Ezinne, Dipesh Bhattarai, Kingsley K. Ekemiri, Gabrielle N. Harbajan, Anesha C. Crooks, Alex A. Ilechie, Ferial M. Zeried, Uchechukwu L. Osuagwu.

**Project administration:** Ngozika E. Ezinne, Uchechukwu L. Osuagwu.

**Software:** Dipesh Bhattarai.

**Supervision:** Ngozika E. Ezinne, Khathutshelo P. Mashige, Alex A. Ilechie, Uchechukwu L. Osuagwu.

**Validation:** Khathutshelo P. Mashige, Ferial M. Zeried.

**Writing – original draft:** Ngozika E. Ezinne, Dipesh Bhattarai, Anesha C. Crooks, Uchechukwu L. Osuagwu.

**Writing – review & editing:** Ngozika E. Ezinne, Dipesh Bhattarai, Kingsley K. Ekemiri, Gabrielle N. Harbajan, Khathutshelo P. Mashige, Alex A. Ilechie, Ferial M. Zeried, Uchechukwu L. Osuagwu.

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
