## [Decision Letter · Decision Letter 0]

3 Mar 2022

PONE-D-22-03935Demographic profiles of contact lens wearers and their association with lens wear characteristics in Trinidad and Tobago: A retrospective studyPLOS ONE

Dear Dr. Osuagwu,

Thank you for submitting your manuscript to PLOS ONE. After careful consideration, we feel that it has merit but does not fully meet PLOS ONE’s publication criteria as it currently stands. Therefore, we invite you to submit a revised version of the manuscript that addresses the points raised during the review process.

We look forward to receiving your revised manuscript.

Kind regards,

Diego García-Ayuso, PhD

Academic Editor

PLOS ONE

Journal Requirements:

Reviewers' comments:

Reviewer's Responses to Questions

**Comments to the Author**

1. Is the manuscript technically sound, and do the data support the conclusions?

Reviewer #1: Partly

Reviewer #2: Yes

2. Has the statistical analysis been performed appropriately and rigorously? 

Reviewer #1: Yes

Reviewer #2: Yes

3. Have the authors made all data underlying the findings in their manuscript fully available?

Reviewer #1: Yes

Reviewer #2: Yes

4. Is the manuscript presented in an intelligible fashion and written in standard English?

Reviewer #1: Yes

Reviewer #2: Yes

5. Review Comments to the Author

Reviewer #1: In this manuscript, the authors studied the demographic profiles of contact lenses wearers and their associations whit purpose of wear, CLs materials, replacement schedule, and modality of wear in Trinidad y Tobago

The results are interesting, but some important things need to be addressed in methodology and text redaction.

In my opinion the biggest limitation is that since the data was collected only from one clinic the authors could not make a valid inference from their sample for the whole Trinidad y Tobago population.

Line 145: The authors should define the meaning of the different purpose of wear, particularly fashion and therapeutic/cosmesis.

Table 1 and Line 171. Cosmesis is not report in the table 1. The authors should always use the same terminology to make the reading of the work simpler and avoid confusing the reader.

Line 176-177:

“Most (61%) record cards did not reflect the type of lens system used.” Insert lens care system.

“…used multipurpose solution and 14% used Boston solution”. The authors should define which type of lens care is “Boston solution” because Boston solution is a brand of Bausch and Lomb company which includes many different types of solutions.

The authors should include the company data in the manuscript

Minor issue:

Lines 92 and 95: Trinidad y Tobago had been previously defined, authors should use the abbreviations

Reviewer #2: our study provided baseline data on the demographical profile of

269 CL in T&T and could be used for comparison across the Caribbean in future studies.

As stated in the methods section the study is a retrospective study that reviewed Cl wearers clinical records of 243 CL wearers attending a university clinic. However, in the abstract authors should not refer to the record as “About half of the respondents wear” . as there is now acquired responses from the wearers. (lines 40-41)

Lines 140-141

Exclusion criteria, why to exclude contact lens prescribed outside UWI optometry clinic. As the aim of the study is to study characteristic of contact lenses in the population rather than single clinic?

The author mentioned a brand of boston solution, can more details add is it RGP solution? ANY OTHER DETAILS.

Table 1 for better readability and comparison to include number of participants and percentage in one column n (%)

Table 2 contact lens type the sum of percentages does not give a total of 100%

The discussion could include more elaboration on the نثغ findings mentioned in the result section,

Age, gender and employment status

Line 263 “questionnaire-based studies”, the study is retrospective no mention of any questionnaire or patients interview

Conclusion: respondent word should not be used in the study as it is retrospective based

6. PLOS authors have the option to publish the peer review history of their article (what does this mean?). If published, this will include your full peer review and any attached files.

Reviewer #1: No

Reviewer #2: **Yes: **May M Bakkar

---

## [Author Response · Author response to Decision Letter 0]

13 Apr 2022

Response to reviewers’ comments 

Thank you for the very useful comments. We have revised the manuscript according to the constructive comments made by the reviewers. Please find below a point-by-point response to the reviewers’ comments. 

Reviewer 1

1. Reviewers’ comment 

Line 145: The authors should define the meaning of the different purpose of wear, particularly fashion and therapeutic/cosmesis.

Response 

Done. The meanings of fashion and therapeutic/cosmesis have been defined in the manuscript (line 144 to 147). 

2. Reviewers’ comment

Table 1 and Line 171. Cosmesis is not report in the table 1. The authors should always use the same terminology to make the reading of the work simpler and avoid confusing the reader. 

Response

Cosmesis and fashion are basically the same thing in this contest. To avoid confusion, we have used fashion as it best represents the responses. This has been corrected in the table and the entire manuscript as suggested.

3. Reviewers’ comment 

Line 176-177:

“Most (61%) record cards did not reflect the type of lens system used.” Insert lens care system.

Response 

Lens care system has been inserted as suggested (line 179)

4. Reviewers’ comments

The authors should define which type of lens care is “Boston solution” because Boston solution is a brand of Bausch and Lomb Company which includes many different types of solutions.

The authors should include the company data in the manuscript

Response The type of Boston solution used has been defined in the manuscript (line 181 to 182). It now reads: 

‘Most (61%) record cards did not reflect the type of lens care system used. About one-quarter (24.3%) used multipurpose solution [e.gs include Bausch and Lomb Renu advance formula, Renu fresh, Renu sensitive and Biotrue multipurpose solutions used for soft CLs] and 14% used Boston solution by Boston Advance® care system (Bausch & Lomb, Rochester, New York)’. See also footnote in Table 1

5. Reviewers’ comment

Lines 92 and 95: Trinidad y Tobago had been previously defined; authors should use the abbreviations.

Response 

Trinidad and Tobago have been changed to T & T in the entire manuscript as suggested 

Reviewers’ #2 

6. Reviewers’ comment: As stated in the methods section the study is a retrospective study that reviewed Cl wearers’ clinical records of 243 CL wearers attending a university clinic. However, in the abstract authors should not refer to the record as “About half of the respondents wear”. as there is now acquired responses from the wearers. (lines 40-41).

Response 

The statement has been corrected to read’’ About half of the CL wearers used them for fashion (more among those aged 18 to 30 years, 61.0%), therapeutic (more among those <18 years, 43.8%, P = 0.001) and refractive error correction purposes (more in those >40 years, P =0.001)’’ (line 45 to 47) 

We have also replaced “respondents” with “CL wearers” in the manuscript as suggested.

7. Reviewers’ comment 

Lines 140-141

Exclusion criteria, why to exclude contact lens prescribed outside UWI optometry clinic. As the aim of the study is to study characteristic of contact lenses in the population rather than single clinic.

Response 

This was a retrospective study involving records of patients that attended the UWI optometry clinic. For this reason, those who CLs were not prescribed at the UWI clinic were excluded because the information regarding their CLs and other variables needed in this study cannot be verified. 

8. Reviewers’ comment 

The author mentioned a brand of Boston solution, can more details add is it RGP solution? ANY OTHER DETAILS.

Response 

More details about the type of Boston lens care solution used have been added. The section now reads: 

“Boston solution by Boston Advance® care system (Bausch & Lomb, Rochester, New York) (line 180 to 181)”. See also footnote in Table 1

9. Reviewers’ comment

Table 1 for better readability and comparison to include number of participants and percentage in one column n (%). 

Response 

The number of participants and percentage have been put in one column as suggested (see Table 1) and footnote added.

10. Reviewers’ comment 

Table 1 CL type the sum of percentages does not give a total of 100%

Response 

The percentage has been corrected.

11. Reviewers’ comment

The discussion could include more elaboration on the findings mentioned in the result section,

Age, gender, and employment status

Response 

The discussion has been elaborated to include age, gender, and employment status in the manuscript (line 240 to 271).

12. Reviewers’ comment 

Line 263 “questionnaire-based studies”, the study is retrospective no mention of any questionnaire or patients’ interview.

Response 

Retrospective studies include review of patient record as well and does not always have to be surveys or interviews. This has been reflected more across the manuscript. Some of the relevant sections were:

“This was a retrospective review of data from patients prescribed with CL” (Line 134) 

And

This retrospective study investigated the characteristics of CL wearers and their association with lens characteristics using data (line 234 to 235).

13. Reviewers’ comment 

Conclusion: respondent word should not be used in the study as it is retrospective based

Response 

We have replaced “respondents” with “CL wearers” in the manuscript as suggested and have also rephrased the conclusion.

---

## [Decision Letter · Decision Letter 1]

11 Jul 2022

Demographic profiles of contact lens wearers and their association with lens wear characteristics in Trinidad and Tobago: A retrospective study

PONE-D-22-03935R1

Dear Dr. Osuagwu,

We’re pleased to inform you that your manuscript has been judged scientifically suitable for publication and will be formally accepted for publication once it meets all outstanding technical requirements.

Kind regards,

Diego García-Ayuso, PhD

Academic Editor

PLOS ONE

Additional Editor Comments (optional):

Reviewers' comments:

Reviewer's Responses to Questions

**Comments to the Author**

1. If the authors have adequately addressed your comments raised in a previous round of review and you feel that this manuscript is now acceptable for publication, you may indicate that here to bypass the “Comments to the Author” section, enter your conflict of interest statement in the “Confidential to Editor” section, and submit your "Accept" recommendation.

Reviewer #1: All comments have been addressed

Reviewer #2: All comments have been addressed

2. Is the manuscript technically sound, and do the data support the conclusions?

Reviewer #1: Yes

Reviewer #2: Yes

3. Has the statistical analysis been performed appropriately and rigorously? 

Reviewer #1: Yes

Reviewer #2: Yes

4. Have the authors made all data underlying the findings in their manuscript fully available?

Reviewer #1: Yes

Reviewer #2: Yes

5. Is the manuscript presented in an intelligible fashion and written in standard English?

Reviewer #1: Yes

Reviewer #2: Yes

6. Review Comments to the Author

Reviewer #1: The changes made by the authors have improved the manuscript, especially in the sections on methodology, discussion and Limitations and strengths.

Reviewer #2: Thanks for considering all comments. The authors gave satisfying revision and addressed all the comments.

7. PLOS authors have the option to publish the peer review history of their article (what does this mean?). If published, this will include your full peer review and any attached files.

Reviewer #1: No

Reviewer #2: **Yes: **May M Bakkar

---

## [Editor Report · Acceptance letter]

13 Jul 2022

PONE-D-22-03935R1 

Demographic profiles of contact lens wearers and their association with lens wear characteristics in Trinidad and Tobago: A retrospective study 

Dear Dr. Osuagwu:

I'm pleased to inform you that your manuscript has been deemed suitable for publication in PLOS ONE. Congratulations! Your manuscript is now with our production department. 

Kind regards, 

on behalf of

Dr. Diego García-Ayuso 

Academic Editor

PLOS ONE